# TGFβ1 Suppressed Matrix Mineralization of Osteoblasts Differentiation by Regulating SMURF1–C/EBPβ–DKK1 Axis

**DOI:** 10.3390/ijms21249771

**Published:** 2020-12-21

**Authors:** Bora Nam, Hyosun Park, Young Lim Lee, Younseo Oh, Jinsung Park, So Yeon Kim, Subin Weon, Sung Hoon Choi, Jae-Hyuk Yang, Sungsin Jo, Tae-Hwan Kim

**Affiliations:** 1Institute for Rheumatology Research, Hanyang University, Seoul 04763, Korea; bora871011@gmail.com (B.N.); hyosun1988@naver.com (H.P.); mylime20@gmail.com (Y.L.L.); epris12@naver.com (Y.O.); ddochi0501@gmail.com (J.P.); rlath109@naver.com (S.Y.K.); tnqls2808@gmail.com (S.W.); 2Department of Rheumatology, Hanyang University Hospital for Rheumatic Diseases, Seoul 04763, Korea; 3Department of Translational Medicine, Graduate School of Biomedical Science and Engineering, Hanyang University, Seoul 04763, Korea; 4Department of Orthopedic Surgery, Hanyang University Seoul Hospital, Seoul 04763, Korea; spineshchoi@gmail.com; 5Department of Orthopedic Surgery, Hanyang University Guri Hospital, Guri 11923, Korea; jaekorea@hotmail.com

**Keywords:** osteoblast differentiation, mineralization, TGFβ1, SMURF1, C/EBPβ, DKK1

## Abstract

Transforming growth factor β1 (TGFβ1) is a major mediator in the modulation of osteoblast differentiation. However, the underlying molecular mechanism is still not fully understood. Here, we show that TGFβ1 has a dual stage-dependent role in osteoblast differentiation; TGFβ1 induced matrix maturation but inhibited matrix mineralization. We discovered the underlying mechanism of the TGFβ1 inhibitory role in mineralization using human osteoprogenitors. In particular, the matrix mineralization-related genes of osteoblasts such as osteocalcin (OCN), Dickkopf 1 (DKK1), and CCAAT/enhancer-binding protein beta (C/EBPβ) were dramatically suppressed by TGFβ1 treatment. The suppressive effects of TGFβ1 were reversed with anti-TGFβ1 treatment. Mechanically, TGFβ1 decreased protein levels of C/EBPβ without changing mRNA levels and reduced both mRNA and protein levels of DKK1. The degradation of the C/EBPβ protein by TGFβ1 was dependent on the ubiquitin–proteasome pathway. TGFβ1 degraded the C/EBPβ protein by inducing the expression of the E3 ubiquitin ligase Smad ubiquitin regulatory factor 1 (SMURF1) at the transcript level, thereby reducing the C/EBPβ-DKK1 regulatory mechanism. Collectively, our findings suggest that TGFβ1 suppressed the matrix mineralization of osteoblast differentiation by regulating the SMURF1-C/EBPβ-DKK1 axis.

## 1. Introduction

Bone is a living tissue that undergoes constant degeneration and rebuilding throughout life. This bone remodeling is orchestrated by a dynamic relationship between specialized cells and their mediators [1]. Dysregulation of these coordinated processes can lead to bone diseases such as osteoporosis, renal osteodystrophy, and Paget’s disease [2]. Despite the importance of researching bone metabolism to improve the understanding of bone disease pathogenesis, the molecular mechanisms underlying bone remodeling are largely unknown.

In the complex crosstalk in bone remodeling, osteoblasts, differentiated from osteoprogenitors, play a key role in bone formation by synthesizing new collagenous extracellular matrix and then inducing mineralization [1]. The specific markers for the differentiation stages of osteoblast differentiation, alkaline phosphates (ALP), type 1 collagen (COL1), and osteonectin (ON), and osteocalcin (OCN), are expressed at a high level in extracellular matrix maturation and mineralization, respectively [3,4,5].

Interestingly, C/EBPβ, a critical determinant for osteoblast differentiation as a transcriptional factor, is known to regulate receptor activator of nuclear factor-κB ligand (RANKL) or Runt-related transcription factor 2 (RUNX2) expression in osteoblasts as well as induce the expression of OCN by directly binding within its promoter [6,7,8]. Wnt/β-catenin is also a key regulator of osteoblast differentiation and is regulated by several inhibitors including Dickkopf (DKK) [9]. However, a previous study has shown that DKK2 has a role in matrix mineralization rather than only playing a negative role in the Wnt/β-catenin signaling pathway [10]. Moreover, we have reported that the expression of C/EBPβ and DKK1 mediated by vitamin D3 stimulation is required for the mineralization process of osteoblast differentiation [11].

One of the other major mediators that modulates osteoblast differentiation is the transforming growth factor-beta 1 (TGFβ1). TGFβ1 had been considered simply as a bone growth stimulant in some early studies [12,13]. However, accumulated data have shown variable results of TGFβ1 on bone metabolism [14]. Therefore, it has been widely accepted that TGFβ1 has a broad range of effects on bone metabolism: TGFβ1 acts in different roles according to the different types of osteoblastic lineage cells or different stages of bone formation. In particular, TGFβ1 promotes osteoprogenitor proliferation and early differentiation through the Smad2/3 pathways, whereas TGFβ1 inhibits mineralization in the late stage of osteoblast differentiation by the degradation of TGFβ type I receptor via the induction of Smad ubiquitination regulatory factor (SMURF1) 1 and SMURF2 [15,16,17,18]. However, the underlying mechanism is not fully understood.

Here, we used human osteoprogenitors to demonstrate that TGFβ1 suppressed the mineralization of osteoblast differentiation by regulating the SMURF1-C/EBPβ-DKK1 axis. Considering that the misregulation of the TGFβ1 signaling pathways is frequently associated with various human diseases [19], and several diseases or pathologic conditions can lead to secondary osteoporosis or abnormal bone formation [2], we might provide a better understanding of the link between them, as well as new insights into the molecular mechanisms underlying normal bone remodeling.

## 2. Results

### 2.1. TGFβ1 Suppressed Extracellular Matrix Mineralization of Osteoblast Differentiation, but Not Matrix Maturation

Firstly, we optimized the osteoprogenitors treatment of various TGFβ1 dose for 3 days to determine the optimal dose of TGFβ1 in our system. Cell viability and toxicity were investigated, and then we decided to treat 10 ng/mL of TGFβ1 (Appendix A). To determine the effect of TGFβ1 on human osteoblast differentiation, human osteoprogenitors were induced to mature osteoblast with differentiation conditional medium in the presence of TGFβ1 continually for 21 days. As shown in Figure 1A–C, we observed gradual and time-dependent increases of ALP and collagen deposits until 14 days, indicating that TGFβ1 promotes extracellular matrix maturation in the osteoblast differentiation stage (0–14 days). However, in the late stage (14–21 days), extracellular matrix mineralization was suppressed by TGFβ1 treatment as indicated by the weaker intensity of Alizarin red staining (ARS), Von Kossa, and hydroxyapatite (HA) staining and lower ARS concentration than those of the vehicle controls (Figure 1D–F). Moreover, treatment with TGFβ1 in osteoprogenitors significantly upregulated SBE4 and ALP promoter activities, but it downregulated the OCN promoter in a dose-dependent manner and had no effect on BSP (Figure 1G). Collectively, these data suggest that TGFβ1 promotes extracellular matrix maturation in the early stage of osteoblast differentiation while inhibiting extracellular matrix mineralization in the late stage.

### 2.2. TGFβ1 Downregulated C/EBPβ and DKK1 Expression during Osteoblast Differentiation

Previously, we reported that C/EBPβ and DKK1 play critical regulatory roles in the matrix mineralization of osteoblast differentiation [11]. Therefore, we investigated the protein levels of C/EBPβ and DKK1 as well as other well-known osteoblast markers such as RUNX2 and OCN; extracellular matrix mineralization marker and ON; and extracellular matrix maturation marker. As expected, the protein levels of C/EBPβ, DKK1, RUNX2, and OCN were gradually increased in a time-dependent manner during osteoblast differentiation, whereas the protein levels of C/EBPβ, DKK1, RUNX2, and OCN were decreased by TGFβ1 treatment (Figure 2A). Protein and mRNA expression showed that TGFβ1 downregulated OCN but upregulated ON and COL1A during osteoblast differentiation, indicating that TGFβ1 sustained the matrix maturation stage but inhibited matrix mineralization. To further explore the physiological differences in osteoblast differentiation, we showed that TGFβ1 increased and sustained ON expression as a matrix maturation marker during differentiation but inhibited OCN expression as a matrix mineralization marker using immunofluorescence (Figure 2B). Taken together, we identified the effect of TGFβ1 on the molecular mechanism of osteoblast differentiation: TGFβ1 promoted the extracellular matrix maturation via upregulating ON and suppressed extracellular matrix mineralization via downregulating C/EBPβ, DKK1, and OCN (Figure 2C).

### 2.3. Anti-TGFβ1 Antibody Reversed TGFβ1-Mediated Suppression of Mineralization

Next, we performed additional experiments using anti-TGFβ1 antibody. Consistent with the above results, the addition of TGFβ1 suppressed mineralization during osteoblasts differentiation. Anti-TGFβ1 treatment rescued the suppressive effect of TGFβ1 on mineralization (Figure 3A). TGFβ1 induced expressions of ALP and COL1, but anti-TGFβ1 blocked these effects (Figure 3B,C). As shown in Figure 3C, qPCR results showed that anti-TGFβ1 antibody reversed the TGFβ1-mediated reductions of C/EBPβ and DKK1 expression.

### 2.4. TGFβ1 Inhibited DKK1 Expression by Inducing Ubiquitination of C/EBPβ Protein

To identify the mechanism underlying the downregulation of C/EBPβ and DKK1 by TGFβ1 in osteoblast differentiation, we treated osteoprogenitors with TGFβ1 and investigated the expression of C/EBPβ and DKK1 at both the protein and gene levels. TGFβ1 decreased the protein level of C/EBPβ and DKK1 dose-dependently (Figure 4A). Interestingly, the mRNA level of C/EBPβ did not significantly change in the presence of TGFβ1, while that of DKK1 decreased (Figure 4B). In the presence of Actinomycin D (Actino. D), a transcription blocking agent, there were no significant differences in the C/EBPβ mRNA expression level with and without TGFβ1 treatment (Appendix A). We next verified the differences in the cellular mechanisms regulating C/EBPβ and DKK1 protein synthesis using cycloheximide (CHX), a protein synthesis inhibitor, and MG132, a proteasome degradation inhibitor. CHX treatment led to a significant decrease in the protein levels of C/EBPβ and DKK1 in the presence of TGFβ1 compared to vehicle controls (Figure 4C). Exposure to TGFβ1 effectively reduced the C/EBPβ protein but accumulated the C/EBPβ protein in MG132-treated cells (Figure 4D). Interestingly, TGFβ1 triggered the overall ubiquitination of both endogenous and exogenous C/EBPβ protein (Figure 4E,F). The reduction of exogenous C/EBPβ protein in 293T cells by TGFβ was consistent with our results of the alteration of endogenous C/EBPβ protein in osteoprogenitors (Appendix A). Notably, we observed that TGFβ1 induced poly-ubiquitinated C/EBPβ protein, suggesting that the downregulation of C/EBPβ-DKK1 by TGFβ1 might be controlled by post-translational mechanisms of C/EBPβ.

### 2.5. TGFβ1 Induced SMURF1 to Regulate C/EBPβ-DKK1 Expressions

Since SMURF expression in osteoblasts was induced by TGFβ1 and the expression of SMURF1 was responsible for osteoblastic activity and bone formation, we confirmed alterations in expression after TGFβ1 treatment [20,21]. Upon TGFβ1 treatment, both mRNA and protein expression of SMURF1 were induced and altered the expression levels of TGFβ-inducible proteins such as phosphor (p)-smad2, p-smad3, and ALK5 in osteoprogenitors (Figure 5A,B). The induction of SMURF1 promoter by TGFβ1 treatment also were observed (Appendix A). To investigate whether SMURF1 regulates the C/EBPβ protein in osteoprogenitors, we investigated the SMURF1 knockdown osteoprogenitor using two types of SMURF1 siRNA mixed in further experiments (Appendix A). Intriguingly, SMURF1 knockdown (siSMURF1) and TGFβ1 specific inhibitor (SB431542) were sufficient to attenuate TGFβ1-mediated C/EBPβ and DKK1 protein degradation (Figure 5C,D). These results suggest that the upregulation of SMURF1 by TGFβ1 might overlap with previously known smad2/3-related mechanisms and could associate with C/EBPβ protein degradation.

## 3. Discussion

In the present study, we replicated a previously reported finding that the C/EBPβ-DKK1 axis may play a dual stage-dependent role in osteoblast differentiation: C/EBPβ and DKK1 expression levels are decreased in extracellular matrix maturation but increased in the mineralization phase of osteoblast differentiation [11]. Conversely but similarly, TGFβ1 induced ALP and collagen synthesis in osteoprogenitors during the matrix maturation stage but suppressed mineralization. Moreover, we discovered the underlying mechanism of the inhibitory role of TGFβ1 in mineralization. It was accompanied by the downregulation of DKK1 following the degradation of the C/EBPβ protein by the ubiquitin–proteasome pathway via inducing SMURF1 expression. Collectively, TGFβ1 suppressed the mineralization of osteoblasts differentiation by regulating the SMURF1–C/EBPβ–DKK1 axis.

TGFβ1 is a pleiotropic cytokine ubiquitously expressed in most tissues. Since TGFβ1 was first identified in 1983 [22], accumulated data and experience have established its regulatory role in a broad range of biological processes including immune responses, angiogenesis, and bone formation [16,23,24]. Parallel with the wide range of roles, the dysregulation of TGFβ1 expression and activity contributes to a number of disease states, including many cancers, cardiovascular disease, and musculoskeletal disease [19]. Additionally, the bifunctional and context-dependent nature of TGFβ has also been documented [25]. In bone metabolism, we reported previously that TGFβ1 has both inhibitory and stimulatory effects on osteoclast differentiation mediated by SMAD1 and 3 signaling [26].

We investigated the regulatory role of TGFβ1 in osteoblast differentiation. Not surprisingly but interestingly, TGFβ treatment for the first 3 days significantly promoted ALP expression and collagen synthesis in the matrix maturation phase. Moreover, this early and transient exposure of TGFβ1 exhibited a subsequent increase in mineralization (Data not shown). However, continuous exposure to TGFβ1 suppressed bone mineralization. These results are consistent with previous studies [15,18,20]. Data from several in vitro experiments have reported the dual, stage-dependent role of TGFβ1 on osteoblast differentiation. TGFβ1 increases bone formation mainly by recruiting progenitors and stimulating their proliferation, and promoting the early stages of differentiation (bone matrix maturation). Conversely, it blocks the later phases of differentiation and mineralization [15,18,27]. These data derived from in vitro studies might reflect in vivo responses of the bone remodeling mechanism; TGFβ1 serum concentrations significantly increased during the early healing period in patients with bone fracture, and they decreased continuously during the late healing period [28].

Moreover, TGFβ1 appeared to act differently according to doses even in the same stage. In the early stage of osteoblast differentiation, ALP activity significantly increased with increasing doses of TGFβ1 (1 ng/mL to 10 ng/mL). However, cell death occurred when the TGFβ1 dose was increased even higher, over 50 ng/mL (Appendix A). Previous studies have resulted in similar data regarding the effect of TGFβ1 on ALP activity using human osteoblasts, but there were controversies in the optimal doses of TGFβ1. ALP activity markedly increased with 1 and 2 ng/mL TGFβ1 treatment compared to negative control and 0.5 and >5 ng/mL TGFβ1 treatment [29]. These variable results might depend on cell density and the timing of TGFβ1 treatment.

To focus on the pathologic mechanism of TGFβ1 on the inhibition of mineralization, we observed that continuous TGFβ1 treatment induced the retention of the matrix maturation phase by sustaining collagen production, leading to impaired mineralization. A similar mechanism was observed with Activin A, which is a member of the TGFβ superfamily. Activin A has an inhibitory role in bone formation by altering the extracellular matrix composition such as ALP and collagen [30] or without ALP and collagen. [30,31]. Furthermore, OCN is known to be a marker for bone mineralization, but it increased in bone formation in OCN knockout mice [32]. However, two recent independent studies showed the functional role of OCN; OCN knockout mouse exhibits a reduction of bone mineralization, strength, and its related metabolism [33,34]. These results support our conclusion that TGFβ1 diminished the process of bone mineralization induced by osteoblasts through OCN reduction. Further experiments are needed to demonstrate that TGFβ1 inhibited OCN expression during osteoblast differentiation.

With regard to molecular mechanisms, our results indicate that the effects of TGFβ1 on bone mineralization may involve DKK1, which is a secreted protein originally located within the endoplasmic reticulum (ER) in the cytoplasm. Several studies have found that ER stress induces DKK1 secretion, eventually leading to cell apoptosis [35,36,37]. Osteoblasts can either transform when they are embedded in bone as osteocytes or undergo apoptosis for bone mineralization at the end phase of bone formation [38]. Therefore, most osteoblasts become a mineralized component of bone with apoptosis rather than osteocytes, and DKK1 upregulation seems indispensable in achieving the terminal feature of bone. Our data show how TGFβ1 can regulate the mineralization of osteoblast differentiation. TGFβ1 suppressed mineralization by the downregulation of DKK1 following the degradation of the C/EBPβ protein via SMURF1-induced ubiquitination.

The present study has a few limitations. First, the function of TGFβ1-induced ON in bone metabolism is unclear and needs further study. The ON is known for its vital role in bone mineralization [39] but did not alter osteoblast differentiation in our system (Appendix A). Second, we investigated only TGFβ1 among three TGFβ isoforms; TGFβ1, TGFβ2, and TGFβ3. When considering the promoter regions of the genes encoding different TGFβ isoforms [40], different TGFβ isoforms might have different roles in bone metabolism. However, we focused on TGFβ1, because it is the dominant species in human bone [41]. Third, we found that TGFβ1 treatment reduced active β-catenin protein expression in osteoprogenitors during osteoblast differentiation, but the negative regulatory signaling of β-catenin remains unknown. Despite these limitations, our data support the dose stage-dependent role of TGFβ1 in osteoblast differentiation using human osteoprogenitors. We provide novel insight into the molecular mechanisms underlying the inhibitory role of TGFβ1 in mineralization. Given that the components of TGFβ signaling are often found to be deregulated in a variety of bone-related disease, understanding how TGFβ1 works in mineralization is important not only for the fundamental understanding of bone metabolism but also for opening up new opportunities for drug development.

## 4. Materials and Methods

### 4.1. Human Bone Sample Collection

Human bone specimens were collected from patients with primary osteoarthritis (OA) who underwent total knee replacement at the Hanyang University Guri Hospital. Surgical knee bone samples were collected from 25 patients with OA (mean ages 74.32 ± 7.13 years). OA was diagnosed according to the clinical classification criteria developed by the American College of Rheumatology [42], and patients with OA secondary to other diseases including connective tissue diseases were excluded. Surgical spinal bone samples were collected from 9 patients with non-inflammatory spinal disease (mean ages 68.11 ± 9.01 years). The present study was approved by the Institutional Review Board of Hanyang University Guri Hospital (IRB file No. 2018-07-024), Hanyang University Seoul Hospital (IRB file No. 2017-05-003), and was carried out in accordance with the Declaration of Helsinki. All patients provided written informed consent, and all data were de-identified and anonymous.

### 4.2. Isolation of Human Primary Osteoprogenitors and Differentiation of Osteoblasts

Cancellous bones from surgical knee bone samples and spinal bones were cut into small pieces (bone chips) using rongeur and operating scissors. These bone chips were vigorously washed by vortexing for 10 min at least 3 times with phosphate-buffered saline (PBS) containing antibiotics to remove non-adherent bone marrow cells completely. These bone chips were placed in cell culture plates and incubated in Dulbecco’s modified eagle medium (DMEM) growth medium containing 10% fetal bovine serum (FBS) and 1% antibiotics to isolate osteoprogenitors using outgrowth methods [43,44,45]. Osteoprogenitors were stimulated with conditioned medium containing osteogenic supplements including 50 μM ascorbic acid (AA), 10 mM β-glycerolphosphate, and 100 nM dexamethasone to induce osteoblast differentiation, as described in previous studies [46,47,48].

### 4.3. Assessment of Osteoblast Differentiation

We investigated the impact of TGFβ1 on osteoblast differentiation according to the following two stages: matrix maturation (initial stage, 0–7 days) and matrix mineralization (late stage, 14–21 days). After examining various doses of TGFβ1 to find an optimal dose on osteoblast differentiation in our system (Appendix A), TGF-β was continually maintained at a constant concentration (10 ng/mL) throughout culture.

In the matrix maturation stage of osteoblast differentiation (initial stage), we assessed the expression of ALP and collagen. ALP was assessed using an ALP activity colorimetric assay kit (K412; Biovision, Milpitas, CA, USA) and ALP staining (85L2; Sigma-Aldrich, St. Louis, MO, USA). The extent of the collagen deposition was assessed with collagen staining (ab150681; abcam, Cambridge, UK) and quantified with a total collagen assay according to the manufacturers’ instructions (K218–100; Biovision). For the assessment of the matrix mineralization phase (late stage), calcium deposition was visualized using several staining methods including Alizarin red staining (ARS; A5533; Sigma-Aldrich) for calcium deposition, Von Kossa staining using 5% aqueous silver nitrate solution (S7179; Sigma-Aldrich) for calcium phosphate, and hydroxyapatite staining (HA; PA-1503; Lonza, Basel, Switzerland). ARS and HA stains for the mineralization stage were assessed and quantified. For ARS quantification, stained wells were extracted with absolute acetic acid at 37 ℃ for 30 min followed by centrifugation. The supernatant was transferred to a white 96-well plate and read at an excitation wavelength of 405 nm with an ELISA plate reader. For Von Kossa quantification, stained images were analyzed by Image J. For HA quantification, stained cells were read at an excitation wavelength of 492 nm and an emission wavelength of 550 nm.

### 4.4. Total Collagen Assay

Total collagen was assessed with the supernatant of each well according to the manufacturers’ instructions (K218-100, Biovision). By adding 12 M HCl and incubating at 120 °C for 3 h, acidic hydrolysis takes place and hydroxyprolines are formed. After homogenization, the sample was clarified by adding 4 mg of activated charcoal. The hydroxyprolines react in an oxidative reaction to a chromophore with a specific absorbance at 560 nm, correlating to the collagen content of the sample.

### 4.5. Luciferase Assay

293T cells or MG63 were co-transfected with each Smad binding element-4 (SBE4), ALP, bone sialoprotein (BSP), OCN, (1 μg/well), SMURF1 promoter and Renilla (0.5 μg/well) using Lipo3000 (L3000015; Thermo Fisher, Waltham, MA, USA). Two days after transfection, the cells were reseeded, treated with TGFβ1 for 24 h, and lysed to conduct a luciferase assay (E1500; Promega, Madison, WI, USA), according to the manufacturer′s protocol. The luciferase activity was measured with a Luminometer (Berthold, Bad Wildbad, Germany) and normalized to Renilla luciferase.

### 4.6. Constructs, Transfection, and Reagents

Recombinant human TGFβ1 (100-21; Peprotech, Rocky Hill, NJ, USA), MG132 (474791; EMD Millipore, Burlington MA, USA), CHX (C4859; Sigma-Aldrich), Actinomycin D (A9415; Sigma-Aldrich), and TGFβ1 blocker (MAB240-100; R&D Systems, Minneapolis, MN, USA) were obtained. DKK1 cDNA plasmid (HG10170-CY) and empty plasmid (CV013) were purchased from Sino Biological (Wayne, PA, USA). SBE4-Luc (#16495) was purchased from Addgene (Watertown, MA, USA). C/EBPβ and empty vector were kindly given by Dr. Yun Jong Lee (Division of Rheumatology, Department of Internal Medicine, Seoul National University Bundang Hospital, Korea) [49,50]. Bone-related promoters such as ALP, BSP, and OCN were kindly given by Dr. Kwang Yeol Lee (College of Pharmacy, Chonnam National University, Korea) [51]. HA-Ubi plasmid, MG63, and 293T cell lines were kindly given by Dr. Heekyoung Chung (Department of Pathology, College of Medicine, Hanyang University, Korea) [52]. SMURF1 promoter was kindly given by Dr. Jeong-Tae Koh (School of Dentistry, Chonnam National University, Korea) [53]. Small interfering RNA (siRNA) oligos were obtained from Genolution Inc. (Seoul, Korea). The siSMUR1 siRNA oligos were as follows:

siControl: 5′-CCUCGUGCCGUUCCAUCAGGUAGUU-3′;

siSMURF1#1: 5′-CCAGUAUUCUACGGACAAU-3′;

siSMURF1#2: 5′-CAUGAAAUGCUGAAUCCUU-3′;

siSMURF1#3: 5′-CCAGCACUAUGAUCUAUAUUU-3′;

siSMURF1#4: 5′-GUGCCAUGAAAUGCUGAAUUU-3′;

siSMURF1#5: 5′-GUCCGGUUGUAUGUAAACUUU-3′.

The transfection of primary osteoprogenitors was carried out using Lipofectamine 3000 (L3000015, Thermo Fisher) according to the manufacturer’s protocol.

### 4.7. Immunoblot and RT-qPCR

Protein level and RNA expression were analyzed using immunoblotting and RT-qPCR, respectively, with standard and basic methods as used in a previous study [54]. RT-qPCR was performed on a CFX96 Real-Time PCR detection system (BR18B-5200, Bio-Rad Laboratories, Hercules, CA, USA). The expression of each target gene was normalized to GAPDH. Normalized expression values were averaged, and then average fold changes were calculated. Primers used for PCR were as follows: ALP_F-ACGAGCTGAACAGGAACAACGT; ALP_R-CACCAGCAAGAAGAAGCCTTTG; COL1_F-AGTGGTTTGGATGGTGCCAA; COL1_R-GCACCATCATTTCCACGAGC; ON_F-GGATGAGAACAACACCCCCA; ON_R-TTTGCAAGGCCCGATGTAGT; OCN_F-AGCCACCGAGACACCATGAGA; OCN_R-CTCCTGAAAGCCGATGTGGTC; RUNX2_F-GTGGCCTTCAAGGTGGTAG; RUNX2_R-ACTCTTGCCTCGTCCACTC; C/EBPβ_F-CGACGAGTACAAGATCCGGC; C/EBPβ_R-TGCTTGAACAAGTTCCGCAG; DKK1_F-CACACCAAAGGACAAGAAGG; DKK1_R-CAAGACAGACCTTCTCCACA; SMURF1_F-CCCCAGGATACCAAGAGACCT; SMURF1_R-GGCTCCTTGAGTTGGCACT; SMURF2_F-CCTGACAGTACTCTGTGCAAAA; SMURF2_R-ATTGCCCAGATCCATCAACCA.

The primary antibodies for C/EBPβ (sc-7962), DKK1 (sc-374574), ON (sc-73472), and SMURF1 (sc-100616) were from Santa Cruz Biotechnology (Dallas, TX, USA). Non-phospho (Active) β-catenin (Ser45) (19807), β-catenin (9562), RUNX2 (12556), p-SMAD2 (3101), SMAD2 (3122), p-SMAD3 (9523), SMAD3 (9520), SMURF2 (12024), β-actin (4970), and GAPDH (2118) were from Cell Signaling Technology (Danvers, MA, USA). ALK-5 (AF3025) was from R&D Systems (Minneapolis, MN, USA). OCN (ab12320) was from Abcam. Secondary antibodies for goat anti-rabbit (111-035-003) and goat anti-mouse (115-035-003) were obtained from Jackson Immunoresearch (West Grove, PA, USA).

### 4.8. Immunofluorescence (IF)

The differentiated osteoprogenitors were washed twice with PBS and fixed with 10% formalin for 15 min. This was followed by permeabilization with PBS containing 0.1% Triton X-100 and 1% BSA for 1 h and incubation with primary antibody overnight. Then, they were washed with PBS and incubated with Cy3 or Alexa 488-conjugated secondary antibody for 1 h. The stained cells were washed with distilled water and mounted with DAPI (H1200, Vector, Burlingame, CA, USA). To visualize stained cells, immunofluorescence images were obtained using a confocal microscope (Leica Microsystems, Wetzlar, Germany). Antibodies used in IF were as follows: ON (sc-73472; Santa Cruz Biotechnology, Dallas, TX, USA), OCN (sc-365797; Santa Cruz, TX, USA), Alexa-488 (A11001; Thermo Fisher), and Cy3 (A10520, Thermo Fisher).

### 4.9. Immunoprecipitation (IP)

293T cells were co-transfected with HA-Ubi (1 μg) and C/EBPβ (3 μg) and incubated for 24 h, and then treated with TGFβ1 or Vehicle for 8 h. The stimulated cells were lysed by non-SDS lysis buffer (150 mM NaCl, 50 mM Tris pH 7.4, 10% glycerol, and 1% Triton X-100) containing proteinase and phosphatase inhibitors. For IP, C/EBPβ antibody (5 µg) was added to protein lysates (1 µg) and subjected to incubation on a rotator for overnight at 4 °C. The next day, protein A/G agarose beads were added to the mixed lysates to incubate at 4 °C for 1 h and then washed three times with ice-cold lysis buffer containing protease inhibitors. Immunoprecipitated proteins were dissolved in 2× Laemmli sample buffer and analyzed by immunoblotting.

### 4.10. Statistical Analysis

Statistical analysis and the graphical representation of data were performed using GraphPad Prism 6 software. A two-tailed *t*-test was used to compare data between two unpaired groups. Values are presented as mean ± standard deviation (SD) or standard error of the mean (SEM) from at least three independent experiments.

## 5. Conclusions

TGFβ1 induced ALP and collagen synthesis in the extracellular matrix maturation stage but suppressed the mineralization of osteoblast differentiation by the downregulation of DKK1 following the degradation of the C/EBPβ protein via SMURF1-induced ubiquitination.

## Figures and Tables

**Figure 1 ijms-21-09771-f001:**
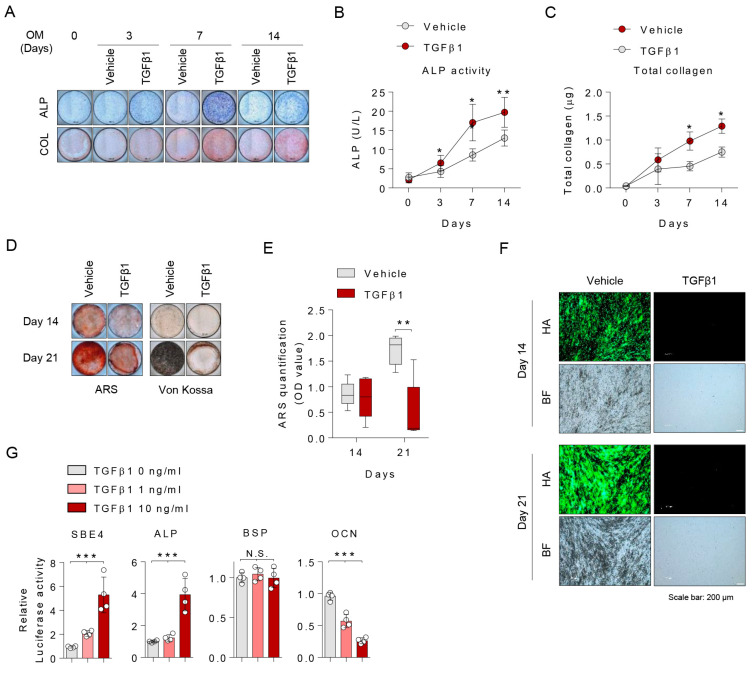
Transforming growth factor β1 (TGFβ1) promotes extracellular matrix maturation in the early stage and suppresses mineralization in the late stage of osteogenic differentiation. Osteoprogenitors were differentiated into osteoblasts and continually stimulated by Vehicle or TGFβ1 (10 ng/mL) during osteoblasts differentiation. At the indicated days, osteogenic differentiation activity was assessed by (**A**) alkaline phosphates (ALP) and collagen (COL) staining; scale bar is 200 μm, (**B**) ALP activity of (**A**), (**C**) total collagen contents of (**A**), (**D**) Alizarin red staining (ARS) and von Kossa staining, (**E**) ARS quantification of (**D**), and (**F**) hydroxyapatite (HA) staining. All images are representative from the five independent experiments. (**G**) 293T cells were transfected with the promoter plasmids indicated for 48 h, treated with 1 and 10 ng/mL TGFβ1 for 24 h, and then analyzed with a luciferase assay (*n* = 4). * *p* < 0.05; ** *p* < 0.01; *** *p* < 0.001; N.S. Not significant. Values are expressed as the mean ± SEM (Student’s two-tailed *t*-test).

**Figure 2 ijms-21-09771-f002:**
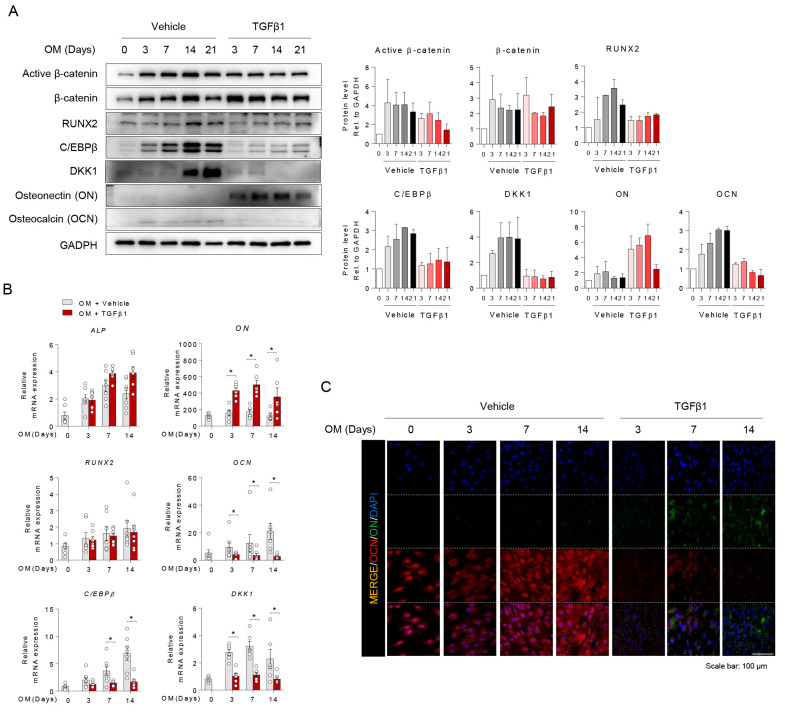
TGFβ1 downregulated CCAAT/enhancer-binding protein beta (C/EBPβ) and Dickkopf 1 (DKK1) expression in osteoblast differentiation. Osteoprogenitors were differentiated into osteoblasts with vehicle or TGFβ1 (10 ng/mL) for the indicated days. At the days indicated, osteogenic differentiation activity was analyzed. (**A**) Immunoblotting with active β-catenin, β-catenin, RUNX2, CCAAT/enhancer-binding protein beta (C/EBPβ), DKK1, ON, OCN, and GAPDH were analyzed and quantified with Image J and normalized to GAPDH. (**B**) RT-qPCR with ALP, COL1A1, ON, OCN, RUNX2, C/EBPβ, and DKK1 were analyzed and normalized to GAPDH (*n* = 5). (**C**) Immunostaining with OCN (red), ON (green), and DAPI (blue) were analyzed and immunofluorescence images are representative from the three independent experiments; scale bar is 100 μm. Values are expressed as the mean ± SD. * *p* < 0.05.

**Figure 3 ijms-21-09771-f003:**
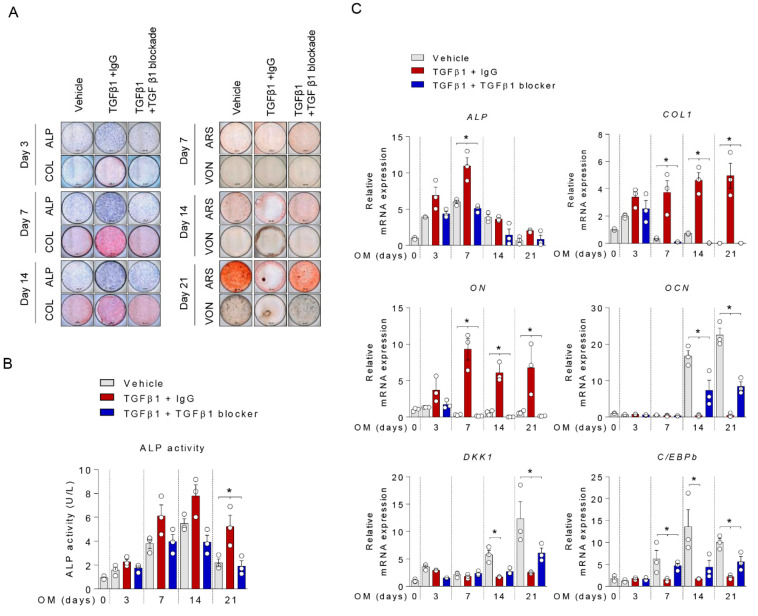
Anti-TGFβ1 antibody blocked the suppressive effect of TGFβ1 on mineralization. Osteoprogenitors were differentiated into osteoblasts in the presence of 10 ng/mL TGFβ1 and Anti-TGFβ1 anbituo for the indicated days and analyzed by (**A**) ALP, COL, ARS, and von Kossa staining; scale bar is 200 μm, (**B**) intracellular ALP activity, and (**C**) the mRNA levels for the osteogenic differentiation-related factors (*n* = 3). All images of (**A**) are representative from the three independent experiments. Values are expressed as the mean ± SD. * *p* < 0.05.

**Figure 4 ijms-21-09771-f004:**
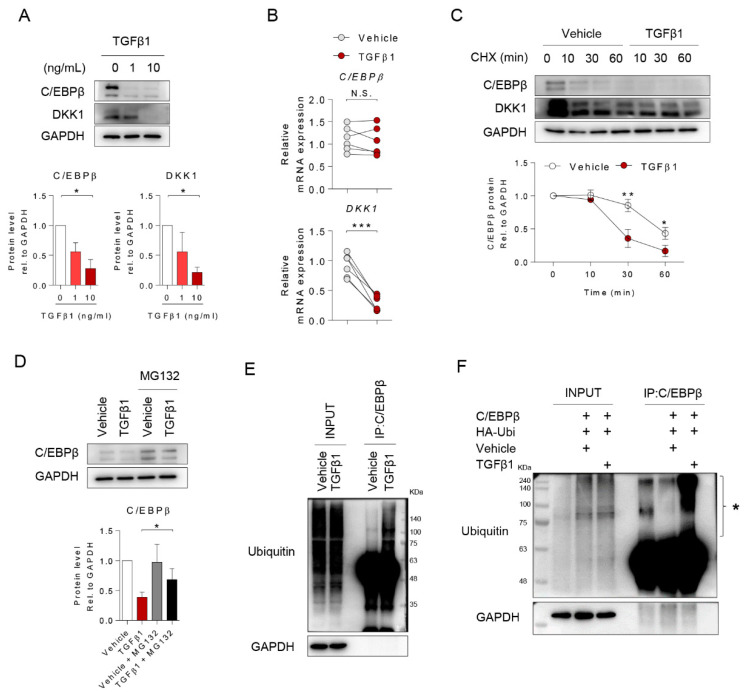
TGFβ1 inhibited DKK1 expressions by ubiquitin-mediated C/EBPβ protein degradation. Osteoprogenitors were stimulated with 0, 1, and 10 ng/mL TGFβ1 for 24 h, lysed, and subjected to analysis for (**A**) immunoblotting (upper) and quantification of the immunoblot images (lower, *n* = 3), and (**B**) qPCR with C/EBPβ and DKK1 expressions (*n* = 5). (**C**) Osteoprogenitors were pretreated with cycloheximide (CHX), protein synthesis inhibitor, for 30 min, followed by treatment with 10 ng/mL TGFβ1 for the times indicated, and analyzed by immunoblotting (upper) and quantification of the immunoblot images (lower, *n* = 5). (**D**) Osteoprogenitors were pretreated with MG132, proteasome inhibitor, for 30 min, followed by treatment with 10 ng/mL TGFβ1 for 24 h and analyzed by immunoblotting and quantified with Image J (*n* = 3; normalized to GAPDH). (**E**) Osteoprogenitors were stimulated with 10 ng/mL TGFβ1 for 24 h, followed by IP with C/EBPβ antibody and analyzed by immunoblotting (*n* = 4). (**F**) 293T cells were co-transfected with C/EBPβ and HA-Ubi plasmids for 24 h, followed by treatment with 10 ng/mL TGFβ1 for 24 h, IP with C/EBPβ antibody, and analyzed by immunoblotting (*n* = 3). Asterisk indicated poly-ubiquitin with C/EBPβ protein. Values are expressed as the mean ± SD. * *p* < 0.05; ** *p* < 0.01; *** *p* < 0.001; N.S. Not significant.

**Figure 5 ijms-21-09771-f005:**
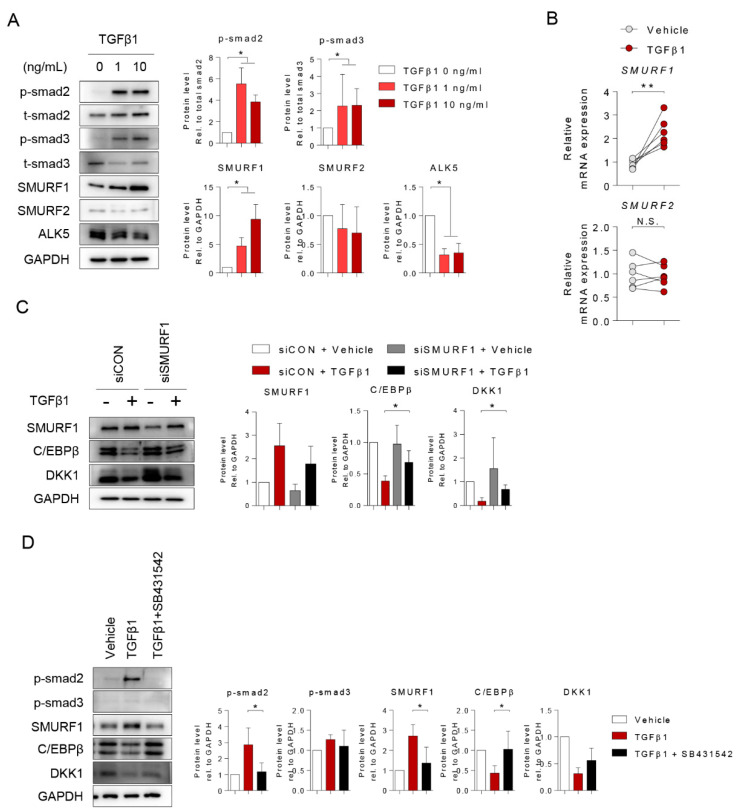
TGFβ1 induced Smad ubiquitin regulatory factor 1 (SMURF1) to regulate C/EBPβ-DKK1 expression. Osteoprogenitors were stimulated with 0, 1, and 10 ng/mL TGFβ1 for 24 h, lysed, and subjected to analysis for (**A**) immunoblotting (left) and quantification of the immunoblot images (right, *n* = 3), and (**B**) qPCR with SMURF1 and SMURF2 expressions (*n* = 6). (**C**) Osteoprogenitors were transfected with sicontrol (siCON) or two siSMURF1 #1 and #2, followed by treatment with 10 ng/mL TGFβ1 for 24 h, and analyzed by immunoblotting and quantified with ImageJ. (**D**) Osteoprogenitors were pretreated with SB411542 for 30 min, followed by treatment with 10 ng/mL TGFβ1 for 24 h, and analyzed by immunoblotting and quantified with ImageJ. Values are expressed as the mean ± SD. * *p* < 0.05; ** *p* < 0.01; N.S. Not significant.

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
