# Peer review of "TGFβ1 Suppressed Matrix Mineralization of Osteoblasts Differentiation by Regulating SMURF1–C/EBPβ–DKK1 Axis"

_ijms, 2020, doi:10.3390/ijms21249771_

Round 1

Reviewer 1 Report

Dear authors,

it is a long time ago that I read such a good manuscript. Congratulations to you and your team.

Therefore, I only have some minor comments:

  • Please explain in the figure legends the abbreviations, eg CHX and MG132 in Figure 4
  • Some figures and the writings in the figures are really small. Please increase the text size in the figures or increase the figures, e.g. Figure 2
  • Figure 3: Did you see no siginificant differences in these parameters? Please check and add significances if there are some.
  • In line 159 to 160 you wrote "in the presence of Actino D...there were no significant differences...". Where can I see the results? In the supplement or are they not shown?

Author Response

Response to the reviewers’ comments

The authors thank the reviewers for very constructive comments on the manuscript (Manuscript ID ijms-1042933) entitled “TGFβ1 suppressed matrix mineralization of osteoblasts differentiation by regulating SMURF1-C/EBPβ-DKK1 axis”. We have adopted nearly all of your suggestions and have provided explanations for reviewers’ questions.

-------------------------------------------------- Reviewer #1 comments---------------------------------------------------

it is a long time ago that I read such a good manuscript. Congratulations to you and your team.

 Therefore, I only have some minor comments:

  • Please explain in the figure legends the abbreviations, eg CHX and MG132 in Figure 4

Response] Thank you for pointing out this. Overall, we added some explanations in revised figure legend.

  • Some figures and the writings in the figures are really small. Please increase the text size in the figures or increase the figures, e.g. Figure 2

Response] Thank you for your comment. We increased the text size in the figures to improve readability.

  • Figure 3: Did you see no siginificant differences in these parameters? Please check and add significances if there are some.

Response] Thank you for your suggestion. We found statistical significance between some parameters, and marked it with asterisks in revised figure 3.

  • In line 159 to 160 you wrote "in the presence of Actino D...there were no significant differences...". Where can I see the results? In the supplement or are they not shown?

Response] Thank you for your comment. The data was shown in Suppl. figure 1. We understand your confusion and decided to cite relevant figure in the RESULT section on page 11 as following:

In the presence of actinomycin D (Actino. D), a transcription blocking agent, there were no significant differences in the C/EBPβ mRNA expression level with and without TGFβ1 treatment (Suppl. Figure 2).

Reviewer 2 Report

The manuscript entitled, "TGFβ1 suppressed matrix mineralization of osteoblasts differentiation by regulating SMURF1-C/EBPβ-DKK1 axis", by Nam et al., have tried to explain the mechanism of the stage dependent activity of TGFβ1 on osteoblast cells.  The efforts by the authors in identifying the role of much studied TGFβ1 signaling in bone cells is appreciating and the manuscript also presents interesting findings. However, the manuscript  could be more interesting and make impact if the following comments could be addressed: 

  1. Authors have mentioned that the osteoprogenitor cells were isolated from the cancellous bone chips. In general, the cell population isolated from bone chips is heterogenous. It contains osteoprogenitors/preosteoblasts, osteoblasts as well as osteocytes. It would be better if the authors have checked the isolated cells were osteoprogenitor rather than mix cells.
  2. The authors have not mentioned if the treatment with TGFβ1 was single or repeated. Some of the earlier findings have reported different actions of TGFβ1 upon single and repeated treatments. It would better if the author could mention the treatment procedure clearly. 
  3. The result (Figure 1) presented shows that the cells were treated with 10ng/ml TGFβ1. How was this dose determined? Did the authors check the response of different doses on the cells or followed the previous literature?
  4. In the lines 112-117, the authors have mentioned time dependent increase in C/EBPβ, DKK1, RUNX2, and OCN and treatment with TGFβ1 decreases C/EBPβ, DKK1 and OCN. However have not mentioned/explained about changes in RUNX2 though it is clear in the Figure 2. It would be better to justify the reduction in Runx2 despite increase in ALP and Col1a1. 
  5. Figure 3A and 3B do not reflect the consistent ALP and COL activities in different conditions. Also could not see consistent with Figure 1. It would be better to mention clearly in the results.
  6. The authors have shown that matrix maturation from 0-14 days and mineralization stage from 14-21 days. It would be interesting also to know if TGFβ1 induces osteoblast proliferation rather than maturation/mineralization in those periods. OR, the status of ALP and COL activities in 14-21 or 14-28 days.
  7. Interestingly, previous studies have explained the positive role of TGFβ1 in bone mineralization in in-vitro as well as in-vivo. So, it would be better if the authors could mention in discussion how their findings are different from previous.

Author Response

Response to the reviewers’ comments

The authors thank the reviewers for very constructive comments on the manuscript (Manuscript ID ijms-1042933) entitled “TGFβ1 suppressed matrix mineralization of osteoblasts differentiation by regulating SMURF1-C/EBPβ-DKK1 axis”. We have adopted nearly all of your suggestions and have provided explanations for reviewers’ questions.

-------------------------------------------------- Reviewer #2 comments---------------------------------------------------

The manuscript entitled, "TGFβ1 suppressed matrix mineralization of osteoblasts differentiation by regulating SMURF1-C/EBPβ-DKK1 axis", by Nam et al., have tried to explain the mechanism of the stage dependent activity of TGFβ1 on osteoblast cells.  The efforts by the authors in identifying the role of much studied TGFβ1 signaling in bone cells is appreciating and the manuscript also presents interesting findings. However, the manuscript could be more interesting and make impact if the following comments could be addressed: 

  1. Authors have mentioned that the osteoprogenitor cells were isolated from the cancellous bone chips. In general, the cell population isolated from bone chips is heterogenous. It contains osteoprogenitors/preosteoblasts, osteoblasts as well as osteocytes. It would be better if the authors have checked the isolated cells were osteoprogenitor rather than mix cells.

Response] Thank you for your constructive comments. We understand your concerns about the identity of bone chip-derived cells. However, a considerable amount of study has used the method of osteoprogenitors isolation from bone chip, and cells obtained by this method have been repeatedly shown to have the traits of the osteoprogenitor [1-6]. Moreover, as shown in Figure 2 A, the protein level of RUNX2 was gradually increased in a time-dependent manner in our system. And as shown in Figure 1 A-C, ALP and collagen were also gradually increased in a time-dependent manner. To be more detailed, ALP activity, a relative early marker of osteoblast differentiation, became obviously increasing after 7 days, indicating that our cells are more immature than osteoblast; osteoprogenitors/preosteoblast. We agree with your idea that there is a possibility that the population of bone-derived cells might be heterogenous mixtures of various osteo-lineage cells. However, in consideration of our results and the accumulated consistent data which have shown in previous studies, we believe that our cells are mainly osteoprogenitors enough to refer them as “osteoprogenitors”.

[1] Wrobel, E.; Leszczynska, J.; Brzoska, E. The Characteristics Of Human Bone-Derived Cells (HBDCS) during osteogenesis in vitro. Cell Mol Biol Lett 2016, 21, 26, doi:10.1186/s11658-016-0027-8.

[2] Yamamoto, T.; Ecarot, B.; Glorieux, F.H. In vivo osteogenic activity of isolated human bone cells. J Bone Miner Res 1991, 6, 45-51, doi:10.1002/jbmr.5650060109.

[3] Jo, S.; Yoon, S.; Lee, S.Y.; Kim, S.Y.; Park, H.; Han, J.; Choi, S.H.; Han, J.S.; Yang, J.H.; Kim, T.H. DKK1 Induced by 1,25D3 Is Required for the Mineralization of Osteoblasts. Cells 2020, 9, doi:10.3390/cells9010236.

[4] Jo, S.; Kang, S.; Han, J.; Choi, S.H.; Park, Y.S.; Sung, I.H.; Kim, T.H. Accelerated osteogenic differentiation of human bone-derived cells in ankylosing spondylitis. J Bone Miner Metab 2018, 36, 307-313, doi:10.1007/s00774-017-0846-3.

[5] Jo, S.; Lee, J.K.; Han, J.; Lee, B.; Kang, S.; Hwang, K.T.; Park, Y.S.; Kim, T.H. Identification and characterization of human bone-derived cells. Biochem Biophys Res Commun 2018, 495, 1257-1263, doi:10.1016/j.bbrc.2017.11.155.

[6] Jo, S.; Wang, S.E.; Lee, Y.L.; Kang, S.; Lee, B.; Han, J.; Sung, I.H.; Park, Y.S.; Bae, S.C.; Kim, T.H. IL-17A induces osteoblast differentiation by activating JAK2/STAT3 in ankylosing spondylitis. Arthritis Res Ther 2018, 20, 115, doi:10.1186/s13075-018-1582-3.

  1. The authors have not mentioned if the treatment with TGFβ1 was single or repeated. Some of the earlier findings have reported different actions of TGFβ1 upon single and repeated treatments. It would better if the author could mention the treatment procedure clearly. 

Response] Thank you for your suggestions. We newly described the TGFβ1 treatment procedure more clearly, as followings:

In the RESULTS section on page 5:

Figure 1. TGFβ1 promotes extracellular matrix maturation in the early stage and suppresses mineralization in the late stage of osteogenic differentiation. Osteoprogenitors were differentiated into osteoblasts and continually stimulated by Vehicle or TGFβ1 (10 ng/ml) during osteoblasts differentiation. At the indicated days, osteogenic differentiation activity was assessed

In the RESULTS section on page 6:

To determine the effect of TGFβ1 on human osteoblast differentiation, human osteoprogenitors were induced to mature osteoblast with differentiation conditional medium in the presence of TGFβ1 continually for 21 days.

In the METHODS section on page 20:

We investigated the impact of TGFβ1 on osteoblast differentiation according to the following two stages; matrix maturation (initial stage, 0~7 days) and matrix mineralization (late stage, 14~21 days). After examining various doses of TGFβ1 to find optimal dose on osteoblast differentiation in our system (Suppl. Figure 1), TGF-β was continually maintained at a constant concentration (10 ng/ml) throughout culture.

  1. The result (Figure 1) presented shows that the cells were treated with 10ng/ml TGFβ1. How was this dose determined? Did the authors check the response of different doses on the cells or followed the previous literature?

Response] Thank you for your comment. To find optimal dose of TGFβ1 on osteoblast differentiation, we examined various doses of TGFβ1 before the main experiment. And as we described in the DISCUSSION section on page 16, we found that TGFβ1 appeared to act differently according to doses even in the same stage; ALP activity significantly increased with increasing dose of TGFβ1 (1 ng/ml to 10 ng/ml), while cell death occurred when the TGFβ1 dose was increased even higher, over 50 ng/ml. Therefore, we decided to treated TGFβ1 dose of 10ng/ml. To provide more information, we decided to describe this as following:

In the RESULTS section on page 6:

Firstly, we optimized the osteoprogenitors treatment of various TGFβ1 dose for 3 days to determine the optimal dose of TGFβ1 in our system. Cell viability and toxicity were investigated , and then we decided to treat 10 ng/ml of TGFβ1. To determine the effect of TGFβ1 on human osteoblast differentiation, human osteoprogenitors were~

In the METHODS section on page 20:

We investigated the impact of TGFβ1 on osteoblast differentiation according to the following two stages; matrix maturation (initial stage, 0~7 days) and matrix mineralization (late stage, 14~21 days). After examining various doses of TGFβ1 to find optimal dose on osteoblast differentiation in our system (Suppl. Figure 1), TGF-β was continually maintained at a constant concentration (10 ng/ml) throughout culture.

  1. In the lines 112-117, the authors have mentioned time dependent increase in C/EBPβ, DKK1, RUNX2, and OCN and treatment with TGFβ1 decreases C/EBPβ, DKK1 and OCN. However have not mentioned/explained about changes in RUNX2 though it is clear in the Figure 2. It would be better to justify the reduction in Runx2 despite increase in ALP and Col1a1. 

Response] Thank you for suggestion. We newly described about changes in RUNX2 in the RESULT section on page 8, as following:

As expected, the protein levels of C/EBPβ, DKK1, RUNX2, and OCN were gradually increased in a time-dependent manner during osteoblast differentiation, whereas the protein levels of C/EBPβ, DKK1, RUNX2, and OCN were decreased by TGFβ1 treatment (Figure 2A).

  1. Figure 3A and 3B do not reflect the consistent ALP and COL activities in different conditions. Also could not see consistent with Figure 1. It would be better to mention clearly in the results.

Response] Thank you for your constructive comment. As you mentioned, we showed both staining and activities of ALP and COL and found relationship between them in Figure 1. In figure 3, difference of staining intensities in different conditions appeared clear. Moreover, it was consistent with mRNA expression (revised Figure 3B). Therefore, we had believed that these were sufficient for supporting our conclusion. However, we agree with your idea that showing data of activity can provide more definite evidences. Therefore, we added available data of ALP activity as revised Figure 3B.  

  1. The authors have shown that matrix maturation from 0-14 days and mineralization stage from 14-21 days. It would be interesting also to know if TGFβ1 induces osteoblast proliferation rather than maturation/mineralization in those periods. OR, the status of ALP and COL activities in 14-21 or 14-28 days.

Response] Thank you for your constructive comment. As you know, we investigated the impact of TGFβ1 on osteoblast differentiation according to distinct osteoblast differentiation phase (matrix maturation and matrix mineralization). Therefore, we had tried to focus only markers representing each stage. And we believed that this strategy can lend strength to conclusion of our study: dual-stage dependent role of TGFβ1, and can improve readability for the readers.

 For these reason, status of early phase marker (ALP, COL) in late phase has not been fully investigated. Only data of ALP and collagen staining in day 14, mRNA expression of ALP and collagen in 14-21 days, and ALP activities in 14-21 days are available in revised figure 3. However, we agree with your idea that it would be interesting and will consider to investigate collagen activity in both early and late stage in further study.

  1. Interestingly, previous studies have explained the positive role of TGFβ1 in bone mineralization in in-vitroas well as in-vivo. So, it would be better if the authors could mention in discussion how their findings are different from previous.

Response] Thank you for your constructive comments. As we described in the INTRODUCION section, TGFβ1 acts highly variable according to the different types of osteoblastic lineage cells or different stages of bone formation due to broad range of TGFβ1 effects on bone metabolism. And as we described in the DISCUSSION section, TGFβ1 can act differently in different dose, and can be affected by cell density and the timing of TGFβ1 treatment, as well as the manner of TGFβ1 treatment; single treatment or repeated treatment.

 Regarding a role of TGFβ1 in bone mineralization, some early studies revealed TGFβ1 as a powerful bone growth stimulant. However, to the best our knowledge, the dual role of TGFβ1 has been widely accepted in these days. This change in our perception of TGFβ1 role in osteoblast differentiation might be results of numerous attempts using variable methods, or due to the development of assessment methods and tools: focusing on bone nodule formation or introducing some novel markers such as collagen type 1, osteonectin, and osteopontin. Even though they became ordinary in these days.

 As described above, we already described that several factors can affect the role of TGFβ1 on bone metabolism. And detailed discussion on results of early studies might blur objective of our study. We decided to briefly introduce some previous studies which revealed the positive role of TGFβ1 in bone mineralization in the INTRODUCTION section on page 4 as following:

One of the other major mediators that modulates osteoblast differentiation is the transforming growth factor-beta 1 (TGFβ1). TGFβ1 had been considered simply as a bone growth stimulant in some early studies [12, 13]. However, accumulated data have shown variable results of TGFβ1 on bone metabolism [14]. Therefore, it has been widely accepted that TGFβ1 has a broad range of effects on bone metabolism: TGFβ1 acts in different roles according to the different types of osteoblastic lineage cells or different stages of bone formation. In particular, TGFβ1 promotes osteoprogenitor proliferation and early differentiation through the Smad2/3 pathways, whereas TGFβ1 inhibits mineralization in the late stage of osteoblast differentiation by the degradation of TGFβ type I receptor via the induction of Smad ubiquitination regulatory factor (SMURF1) 1 and SMURF2 [15-18]. However, the underlying mechanism is not fully understood.

--------------------------------------------------------------------------------------------------------------------------------------

We truly appreciate the time and effort you have given us to share your insightful comments and suggestions. We hope that the revised version of our paper is now suitable for publication in the International Journal of Molecular Sciences.

Round 2

Reviewer 2 Report

The authors have addressed the comments very carefully and with reasonable evidences. I hope the manuscript would be interesting for readers and additional information to previous findings on Roles of TGFβ1 signal in bone cells.